# Oropouche virus outbreaks in northeast Brazil between 2024–25 are characterized by sustained transmission and spread to newly affected areas

Elverson Soares de Melo[1,2*], Sophia Maria Dantas da Silva[2,3],
Gustavo Barbosa de Lima[4], Adalúcia da Silva[4], Alexandre Freitas da Silva[1],
Verônica Gomes da Silva[4], Elisa de Almeida Neves Azevedo[4], Letícia Welter Rother[5],
Keilla Maria Paz e Silva[6], Diego Arruda Falcão[6], Andreza Pâmela Vasconcelos[6],
Mayara Matias de Oliveira Marques da Costa[6], Eduardo Augusto Duque Bezerra[7],
Thiago Franco de Oliveira Carneiro[8], Erik Matthaus de Lima Paiva[8],
Janaina Correia Oliveira[8], Matheus Filgueira Bezerra[9], Marcelo Henrique Santos Paiva[1,2],
Bartolomeu Acioli-Santos[4], Clarice Neuenschwander Lins de Morais[4],
Tulio de Lima Campos[10], Gabriel da Luz Wallau[1,11,12*]

1 Department of Entomology and Bioinformatics Core, Aggeu Magalhães Institute (IAM), Oswaldo Cruz Foundation (Fiocruz/PE), Recife, Brazil, 2 Federal University of Pernambuco (UFPE), Recife, Brazil, 3 Graduate Program in Public Health, Aggeu Magalhães Institute (IAM), Oswaldo Cruz Foundation (Fiocruz/PE), Recife, Brazil, 4 Department of Virology, Aggeu Magalhães Institute (IAM), Oswaldo Cruz Foundation (Fiocruz/PE), Recife, Brazil, 5 Graduate Program in Agronomy, Federal University of Santa Maria (UFSM), Santa Maria, Brazil, 6 Central Laboratory of Public Health of Pernambuco (LACEN-PE), Recife, Brazil, 7 Pernambuco State Department of Health, Recife, Brazil, 8 Central Laboratory of Public Health of Paraíba (LACEN-PB), João Pessoa, Brazil, 9 Department of Microbiology, Aggeu Magalhães Institute (IAM), Oswaldo Cruz Foundation (Fiocruz/PE), Recife, Brazil, 10 Bioinformatics Core, Aggeu Magalhães Institute (IAM), Oswaldo Cruz Foundation (Fiocruz/PE), Recife, Brazil, 11 Department of Arbovirology and Entomology, Bernhard Nocht Institute for Tropical Medicine, WHO Collaborating Center for Arbovirus and Hemorrhagic Fever Reference and Research, National Reference Center for Tropical Infectious Diseases, Hamburg, Germany, 12 Federal University of Santa Maria (UFSM), Santa Maria, Brazil

* elverson.melo@gmail.com (ESM); gabriel.wallau@fiocruz.br (GLW)

## Abstract

Oropouche virus (OROV) has recently expanded in Brazil, establishing transmission in non-endemic regions. This study aims to integrate epidemiological and molecular data to investigate OROV spread in Northeast (NE) Brazil between 2024 and 2025. OROV cases were analyzed regarding ecological risk factors and geographical clustering. Additionally, we sequenced 65 new OROV genomes from the Northeast states of Pernambuco, Paraíba, and Sergipe to infer the virus's spatiotemporal dynamics in NE Brazil. A total of 2,806 confirmed cases were reported between March 2024 and April 2025, affecting 170 municipalities across eight out of nine NE states, with highly heterogeneous incidence. An ecological shift was observed, with OROV transmission moving from Atlantic Forest areas in 2024 to humid Caatinga zones in 2025. Phylogenetic reconstruction revealed multiple independent viral introductions in Northeast in 2024, including two in Pernambuco. The first, originating from the central Amazonas, became the main driver of local transmission and subsequently spread

**Data availability statement:** The sequences generated in this study have been deposited in the GISAID database and are available under the following EPI_SET ID: EPI_SET_260130yq (https://doi.org/10.55876/gis8.260130yq). The alignment used to generate the phylogenetic trees, which is available as supplementary material, also includes sequences produced in this study that were prefixed according to the following convention: hOROV/Brazil/PE-IAM, hOROV/Brazil/PE-LACENPE, hOROV/Brazil/PB-IAM, and hOROV/Brazil/SE-IAM.

**Funding:** This work was supported by the Conselho Nacional de Desenvolvimento Científico e Tecnológico – CNPQ (307209/2023-7 to GLW). The funders had no role in study design, data collection and analysis, decision to publish, or preparation of the manuscript. The authors received no specific funding for this work.

**Competing interests:** The authors declare no conflict of interest.

to Sergipe and Paraíba, causing outbreaks in 2024 and 2025, respectively. The second introduction remained restricted within Pernambuco. While several Northeast municipalities reported high OROV incidence, Jaqueira (Pernambuco) emerged as a key hub for regional viral spread. OROV showed sustained transmission in the region over a two-year period, characterized by marked spatiotemporal displacement consistent with short-lived, rapidly spreading outbreaks, followed by cryptic transmission and subsequent dissemination to new areas, ultimately driving renewed intense outbreaks.

## Author summary

The Oropouche virus, a neglected zoonotic pathogen mostly reported in the Amazon region in the past, has spread and become established in Northeast Brazil between 2024 and 2025. We combined epidemiological information with virus genomes collected from infected people in three states to understand when and where the virus moved. Confirmed cases reached 2,806 across eight of the nine Northeast states, but cases were not evenly distributed: most municipalities reported few cases, while a small number experienced comparatively high incidence. Over time, the outbreak shifted from areas linked to the Atlantic Forest in 2024 to more humid parts of the Caatinga in 2025, showing that transmission can occur in new ecological settings. From the virus genomic information, we found that the Northeast outbreaks were caused by several separate introductions, not a single spread event. Two introductions reached Pernambuco, but one became dominant and seeded outbreaks in neighboring states. Our results indicate that the municipality of Jaqueira, in Pernambuco, was a key source hub for regional spread, and also suggest that the virus likely circulated undetected for weeks to months before large outbreaks were recognized.

## Introduction

The *Orthobunyavirus oropoucheense* (OROV), the causative agent of Oropouche fever, is an arbovirus belonging to the family Peribunyaviridae primarily transmitted by the biting midge *Culicoides paraensis* [1]. First detected in 1955 in Trinidad and Tobago during an outbreak of acute fever among forest workers, it has since become one of the leading causes of arboviral infections in Latin America [2]. Until 2012, OROV was the second most frequent arbovirus infecting humans in Brazil, surpassed only by the dengue virus (*Orthoflavivirus denguei*; DENV). The subsequent emergence of chikungunya virus (*Alphavirus chikungunya;* CHIKV) and Zika virus (Orthoflavivirus zikaense; ZIKV) substantially altered patterns of arbovirus detection and reporting [3]. Nevertheless, OROV has remained epidemiologically relevant and continues to raise public health concerns in Brazil and the region. Since its discovery, there have been more than 30 OROV outbreak events, mostly in South America. In

addition, travel-associated cases have also been identified on other continents, including in Central America (e.g., Cuba) and Europe (Spain, Germany, and Italy) [2,4].

Oropouche fever causes a range of symptoms like high fever, headache, myalgia, and arthralgia, and less frequently, rash, retro-orbital pain, and anorexia. The similarity in clinical symptoms often complicates the differentiation between Oropouche fever and other urban arboviruses circulating in Brazil, such as Dengue, Chikungunya, and Zika. In rare cases, OROV can also reach the central nervous system [5], and there are reports of its potential link to congenital malformations as microcephaly [6]. Lastly, viral RNA has already been detected in multiple fetal tissues following intrauterine death, where OROV was identified as the ethiological agent, with findings suggesting the development of fetus consequences similarly to those caused by Zika virus (ZIKV) infection [7,8].

OROV have a tri partite segmented virus genome having the S – small, M – medium and L – large segments coding for nucleocapsid protein and a non-structural protein, glycoproteins and a non-structural protein, and the polymerase RdRp, respectively [9]. As such, this virus evolves both by antigenic drift, accumulation of non-synonymous mutations that may alter the structure of the viral antigen, and reassortments which occur when two different viral lineages co-infect the same host, giving origin to a lineage with a new segment composition [10]. Phylogenetic reconstruction conducted before the current outbreak (2022-2025) suggested that the OROV genome has undergone various reassortment events over the years. The acquisition of a new M segment usually leads these viruses to encode a different glycoprotein, the virus's main antigenic protein, which can reduce the immune protection conferred by prior OROV infections and change host receptor binding characteristics of the virus [3].

Over the past three years (2023–2025), the Americas have experienced a marked increase in the number of OROV cases. In 2024, 16,239 cases of Oropouche fever were reported in 11 countries, mainly in Brazil (13,785 cases), Peru (1,263 cases), Cuba (626 cases), Bolivia (326 cases) and Colombia (74 cases). In 2025, 12,786 cases (up to July 2025) had already been confirmed across the Americas, with most cases reported in Brazil, Panama, and Peru [11]. In Brazil, this increase of cases was first detected in the Northern region of the country, which includes the states of Amazonas, Acre, Roraima and Rondônia [12]. A new viral lineage was found to be causing this rise in infections. This lineage results from the reassortment between the M segment of viral strains previously circulating in the Northern Region and the L and S segments derived from viruses sampled in neighboring countries such as Colombia, Peru, and Ecuador. It is estimated that this new lineage reemerged around 2013–2014 and circulated for approximately nine years without triggering major human outbreaks, until a 2022–24 wave was detected. This led to the emergence of four distinct OROV clades by early 2024, designated as AMACRO-II, AM-I, AM-II, and AM-III [12]. Subsequently, OROV infection spread to several non-Amazonian Brazilian states, generating a national public health emergency in 2024. Phylogeographic analyses supported evidence of active transmission outside the Northern region and also indicated single or multiple introductions in a number of Brazilian states [13].

The Brazilian Northeast region is the second most populous region of the country (57,244,485 inhabitants) and comprises the highest number of states (Alagoas, Bahia, Ceará, Maranhão, Paraíba, Pernambuco, Piauí, Rio Grande do Norte, and Sergipe), distributed across four different biomes [14]. In this region, two independent OROV introduction events were described, one in Pernambuco and the other in Ceará [13,15,16], and both lineages originated directly from the state of Amazonas. Nevertheless, additional introduction events may have gone undetected due to limited sampling, notably the dynamics underlying the dissemination and maintenance of OROV across states in the Northeast region is still poorly understood. Moreover, the main drivers of OROV sustained transmission in previously non-endemic regions remain to be uncovered. Here, we investigated the epidemiological, spatial, and ecological dynamics of Oropouche fever across Northeast Brazil between 2024 and 2025. We characterized patterns of incidence, spatial clustering, and ecological risk factors associated with OROV transmission at the municipal level. Additionally, we sequenced OROV genomes directly from clinical samples from Northeast states (Pernambuco, Paraíba, and Sergipe) and performed molecular and Bayesian phylogeographic analysis. Our findings uncovered a new introduction event in Pernambuco and revealed that

the Pernambuco outbreak served as a key dissemination hub for the spread of the virus to neighboring states such as Paraíba and Sergipe.

## Methods

### Ethics statement

This study received ethical approval from the Ethics Committee of the Instituto Aggeu Magalhães (CAAE 10117119.6.0000.5190), and the requirement for written informed consent was waived due to anonymity of the participants.

**Epidemiological data collection and spatial analysis.** Data on the number of Oropouche Fever cases per municipality in the Northeast region, along with their corresponding epidemiological reporting weeks, were retrieved from the Oropouche Fever Epidemiological Dashboard maintained by the Brazilian Ministry of Health [17], which compiles records from the e-SUS Sinan surveillance system. To calculate the incidence rate per 100,000 inhabitants, we used the 2024 municipal population estimates provided by the Brazilian Institute of Geography and Statistics (IBGE) [18]. The analysis period spanned from March 2024 to April 2025, corresponding to the first confirmed cases in the region. Information on the case's probable place of infection and municipality of residence was also obtained from the same dashboard. Cases were classified as imported when the reported probable place of infection, as determined by epidemiological surveillance teams based on travel history and exposure, was located in a different state from the individual's residence.

Agricultural production data for the year 2023 were obtained from the IBGE (https://www.ibge.gov.br/estatisticas/economicas/agricultura-e-pecuaria). Spatial data, including the shapefiles of municipal and Intermediate Geographic Regions – IGR (an official territorial unit defined by IBGE in the 2017 regional division of Brazil that articulates clusters of municipalities by linking Immediate Regions (IR) around a higher-order urban center) biomes, climate, vegetation area, and land cover and use of 2024, were also obtained from the IBGE (https://www.ibge.gov.br/geociencias/todos-os-produtos-geociencias.html). We also extracted the most recent data about land cover and use from the MapBiomas project [19]. All shapes and geographic data used in this article are sourced from datasets licensed under Creative Commons Attribution 4.0 (CC BY 4.0), and no proprietary basemaps were utilized.

Municipal incidence rates were summarized by population strata (small: ≤ 50,000; medium: 50,000–200,000; large: > 200,000 inhabitants) using medians and interquartile ranges. Differences across strata were assessed using the Kruskal–Wallis test, with effect size quantified by epsilon-squared ($\varepsilon^2$). Because OROV case counts were sparse and included a large proportion of municipalities with zero reported cases, we assessed the association between population size and incidence using zero-inflated negative binomial regression models fitted to all municipalities. These models account for excess zeros by distinguishing municipalities unlikely to generate cases (structural zeros) from those at risk, while modeling incidence intensity among municipalities with reported cases. We used a population offset to model incidence (cases per 100,000 inhabitants) instead of total case counts. All statistical analyses were conducted in R. Zero-inflated models were fitted using the glmmTMB package with a negative binomial distribution (nbinom2). Statistical significance was evaluated using an α level of 0.05.

To mitigate incidence instability in municipalities with small populations, spatial empirical Bayes smoothing was applied to municipal-level incidence rates using TerraView software (version 4.4) [20]. Geographic data integration and the creation of thematic maps were carried out using QGIS (version 3.40) [21]. Finally, we conducted spatial autocorrelation analysis using GeoDa software [22] (version 1.20). We applied the Empirical Bayes (EB) adjustment for Moran's I as proposed by Assunção and Reis [23]. Global and local Moran's I statistics were calculated to identify clusters of high and low incidence, considering first-order neighbours (queen criterion). Statistical significance was assessed using Monte Carlo permutation tests (n = 999) considering a pseudo p-value <0.05.

**OROV sample collection and whole-genome sequencing.** Initial detection of Oropouche Fever cases was conducted by the Central Public Health Laboratories of Pernambuco (LACEN-PE), Sergipe (LACEN-SE), and Paraíba

(LACEN-PB) using a real-time reverse transcription PCR (RT-qPCR) assay targeting OROV. Serum samples from RT-qPCR–positive patients were sent to the Arbovirus Reference Service at the Aggeu Magalhães Institute (FIOCRUZ Pernambuco). For subsequent molecular analysis, only samples with a Ct (cycle threshold) value below 30 were selected for sequencing.

Viral RNA was extracted using ReliaPrep Viral Total Nucleic Acid Purification Kit (Promega), library preparation was performed with the Illumina COVIDSeq backbone. The amplification of complete OROV genomic segments was performed through the AmpliSeq method, using a set of OROV-specific primer panels, as described by Naveca et al. (2024). High-throughput sequencing was conducted on the MiSeq platform using a 150 bp paired-end approach.

**Genome assembly.** For the reconstruction of the viral genome sequences, sequenced reads were processed using the ViralFlow pipeline (version 1.2.0) [24], which implements a reference-guided genome assembly approach. The reference sequences used correspond to the L (Accession: OL689334.1), M (Accession: OL689333.1), and S (Accession: OL689332.1) segments from the lineage causing the outbreak in Brazil. Samples with less than 70% coverage breadth in any of the three segments were excluded from subsequent analyses.

**Segment-based phylogenetics and recombination filtering.** To perform the evolutionary analysis, in addition to the genomes sequenced in this study, viral genomes from samples collected in other Brazilian localities were also retrieved from public databases NCBI from 2022 up to January 30, 2025, particularly sequences produced by Naveca et. al. (2024), Gräf et al. (2024), Moreira et al. (2024), Lima et al. (2025), and Santos et al. (2025) [12,13,15,25,26]. Each segment was aligned independently using the MAFFT tool [27] (version 7.490), followed by manual curation to remove non-coding sequences and unaligned regions. The RDP5 [28] software was used to detect recombinant regions, and no regions were found.

Phylogenetic trees for each viral genome segment were initially reconstructed using the Maximum Likelihood method implemented in IQ-TREE [29] (version 2.3.6), with evolutionary model selection performed by ModelFinder [30] and applying 1,000 bootstrap replicates. Tree topologies were compared to identify potential phylogenetic incongruities suggestive of new segment reassortment events. As no new reassortments were found, all sequences were used in downstream analysis using concatenated segments.

**Temporal Signal examination.** The three viral segments were concatenated using the SeqKit tool [31], resulting in a representative genome for each sample. As with the individual segments, the concatenated genomes were also aligned using MAFFT, and phylogenetic inference was performed with IQ-TREE. To assess the temporal signal and molecular clock assumption, root-to-tip regression analysis was performed using TempEst [32] (version 1.5.3).

**Phylodynamic and phylogeographical analysis.** To reconstruct the spatiotemporal dynamics of OROV in Northeast Brazil, focusing in particular on municipalities in the states of Pernambuco, Paraíba and Sergipe, we constructed a discrete time-calibrated phylogenetic tree using BEAST v1.10.4 [33]. The evolutionary model applied was GTR + G + I, as recommended by ModelFinder, and codon positions were treated as separate partitions to account for substitution rate variation across sites. An uncorrelated lognormal relaxed molecular clock was employed to allow rate variation among lineages. As a tree prior, we adopted the coalescent Bayesian Skyline model, which allows for the inference of changes in effective population size over time without relying on strict parametric assumptions regarding viral demography. We defined geographic locations as discrete traits for analysis. These included: (i) grouped cluster comprising the three states of Brazil's Southern region (Fig 1A); (ii) a grouped cluster comprising Acre, Rondônia, and southern municipalities of Amazonas; (iii) a set of individual states, Roraima, Rio de Janeiro, Espírito Santo, São Paulo, Amazonas, Ceará, Bahia and Sergipe; and (iv) municipalities reporting OROV cases within Pernambuco and Paraíba, which were analyzed separately due to their local importance in viral dispersion. An asymmetric discrete trait substitution model was applied, combined with a Bayesian Stochastic Search Variable Selection (BSSVS) procedure.

To ensure robustness in the phylogenetic inference, ten independent Markov Chain Monte Carlo (MCMC) runs were conducted using the Metropolis-Hastings algorithm, each comprising 100 million generations. After completion of all

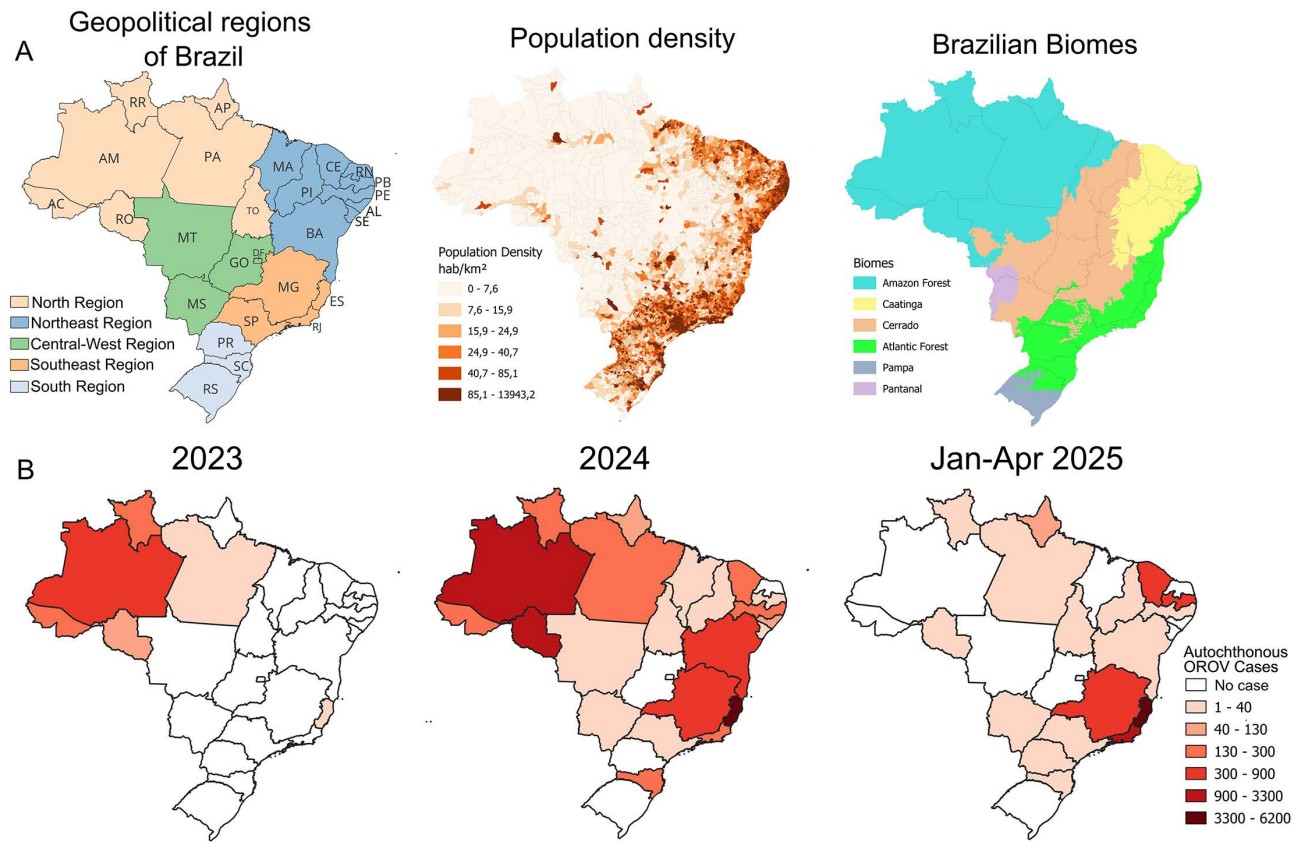

**Fig 1. Geographic and temporal distribution of autochthonous OROV cases across Brazilian states (January 2023 to April 2025) in relation to geopolitical regions, population density and biomes.** (A) The maps illustrate the current geopolitical division of Brazil, highlighting regions and states, population density distribution, and biomes across the country. (B) The maps illustrate the growing geographic spread and increase in the number of autochthonous cases over time, highlighting a notable expansion from the Amazon basin in 2023 to several states in Northeast and Southeast Brazil in 2024 and 2025. Acre (AC), Alagoas (AL), Amapá (AP), Amazonas (AM), Bahia (BA), Ceará (CE), Distrito Federal (DF), Espírito Santo (ES), Goiás (GO), Maranhão (MA), Mato Grosso (MT), Mato Grosso do Sul (MS), Minas Gerais (MG), Pará (PA), Paraíba (PB), Paraná (PR), Pernambuco (PE), Piauí (PI), Rio de Janeiro (RJ), Rio Grande do Norte (RN), Rio Grande do Sul (RS), Rondônia (RO), Roraima (RR), Santa Catarina (SC), São Paulo (SP), Sergipe (SE) e Tocantins (TO). The basemap shapefiles used to produce this figure were obtained from the IBGE municipal mesh dataset (available at: https://www.ibge.gov.br/en/geosciences/territorial-organization/territorial-meshes/2786-np-municipal-mesh/18890-municipal-mesh.html?lang=en-GB) and the Brazilian terrestrial biomes dataset (available at: https://geoftp.ibge.gov.br/informacoes_ambientais/estudos_ambientais/biomas/vetores/Biomas_250mil.zip), both distributed under a CC BY 4.0 license (https://biblioteca.ibge.gov.br/visualizacao/livros/liv102169.pdf).

runs, the chains were combined using *LogCombiner* from the BEAST package, with the initial 10% of states discarded as burn-in. Convergence diagnostics were performed in *Tracer* v1.7.1 [34], by evaluating the Effective Sample Size (ESS) of each parameter. Only parameters with ESS values above 200 were considered reliable, indicating adequate mixing and convergence of the chains. The time to the most recent common ancestor (tMRCA) for clades associated with introductions into Northeast Brazil was estimated using the *TreeStat* tool from the BEAST package. The final Maximum Clade Credibility (MCC) tree was obtained with *TreeAnnotator* (BEAST package) and visualized using *FigTree* (https://github.com/rambaut/figtree/).

Finally, spatial diffusion patterns were assessed and visualized using SpreaD3 (Spatial Phylogenetic Reconstruction of Evolutionary Dynamics) [35], which enabled the spatiotemporal interpretation of viral movement across geographic regions.

## Results

### Epidemiological dynamics of oropouche fever in the Northeast of Brazil

According to the Brazilian Ministry of Health, 834 confirmed cases of Oropouche fever occurred in 2023, most of them in states of the Amazon basin in Northern Brazil (Fig 1). By 2024, this number had increased to approximately 13,800 cases, with autochthonous cases reported in 22 Brazilian states, while by April 2025, around 11,300 cases had already been confirmed, indicating continuous spreading (Fig 1B). Within this epidemiological scenario, the Northeast region of Brazil recorded 1,517 cases of Oropouche fever in 2024 (~11% of 2024 national cases), distributed across 8 states, and 1,319 cases by May 2025, reported in 5 states (~11,4% of 2025 national cases). The sex distribution of cases in the region was approximately balanced, with 50.49% female, 49.33% male, and 0.18% with unreported sex. Regarding age distribution, most cases (62.06%) occurred among individuals aged 20–49 years, followed by those aged 50–69 years (18.97%) and 10–19 years (11.42%); smaller proportions were observed in those aged >70 years (4.80%) and 0–9 years (2.50%). Age was not reported for 0.25% of cases.

Autochthonous transmission in Northeast Brazil was first documented in February, during the eighth epidemiological week (EW) of 2024 (February 18–24), with two cases reported in Maranhão (MA) and one in Bahia (BA) (Fig 2A). In Maranhão, cases were characterized by a scattered distribution between EW 8 and 32. However, a marked increase in the number of cases was observed in Bahia following the initial case along with a small outbreak in Piauí between the 11th and 19th EWs. In Bahia, transmission persisted until the 21st EW(May 19–25), followed by a gradual decline (Fig 2A). Concurrently, an increase in case numbers was observed in Pernambuco (PE), where the first autochthonous case was identified in the EW 16 (April 14–20). The state experienced a peak in infections during the EW 26 (June 23–29), near the end of the rainy season, with the outbreak persisting through the EW 38 (September 15–21). Following the peak of cases in Pernambuco, significant increases were subsequently observed in Alagoas (AL) and Ceará (CE) beginning in early July (at the beginning of the dry season). Although local transmission had already been established in both states by mid-May, the most pronounced growth occurred during the second half of the year. In Ceará, cases continued to be reported through the end of 2024; however, a declining trend was noted starting in the EW 34 (August 18–24), mirroring the pattern observed throughout the Northeast region. Interestingly, this steep decline coincided with the months that have the lowest average precipitation (Fig 2B). Despite this, the first two months of the dry season still allowed sustained transmission in Sergipe (SE), between weeks 25 and 36, and contributed to the continued spread of cases in Alagoas and Ceará.

Although the first autochthonous cases were officially reported in February 2024, initial cases of OROV infection among residents of Brazil's Northeast region were documented between the EW 2 and 5 of 2024 (S1 Fig). Individuals from Pernambuco, Paraíba, Ceará, Maranhão, and Piauí, likely acquired the infection while traveling to the Amazonian region (Fig 2C). However, it was not possible to confirm whether these cases were directly related to local transmission, due to the likely underreporting of early OROV cases at the onset of the outbreak in the Northeast region and the consequent absence of early genomic data from these patients.

The last Northeast state to report autochthonous transmission was Paraíba, with the first locally acquired case confirmed during the 40th EW of 2024 (September 29 – October 5). Only six locally transmitted cases were recorded in 2024; however, a marked increase in incidence was observed beginning in the first EW of January 2025. Case counts peaked at over 165 per week by EW 7 of that year (February 9–15). After this peak, a reducing trend in incidence was observed (Fig 2A). Concurrently with the decline in Paraíba, a new rise in cases emerged in the state of Ceará, along with additional case reports, but with comparatively lower incidence in Bahia and Pernambuco. Particularly for Ceará, where the 2025 outbreak was much larger than in 2024, six municipalities reported cases in both years. Baturité, Aratuba, and Capistrano accounted for most cases in 2025, and Baturité showed the largest increase compared with 2024, with additional cases in Mulungu, Pacoti, and Redenção. Seven other municipalities reported cases only in 2025, and in small numbers.

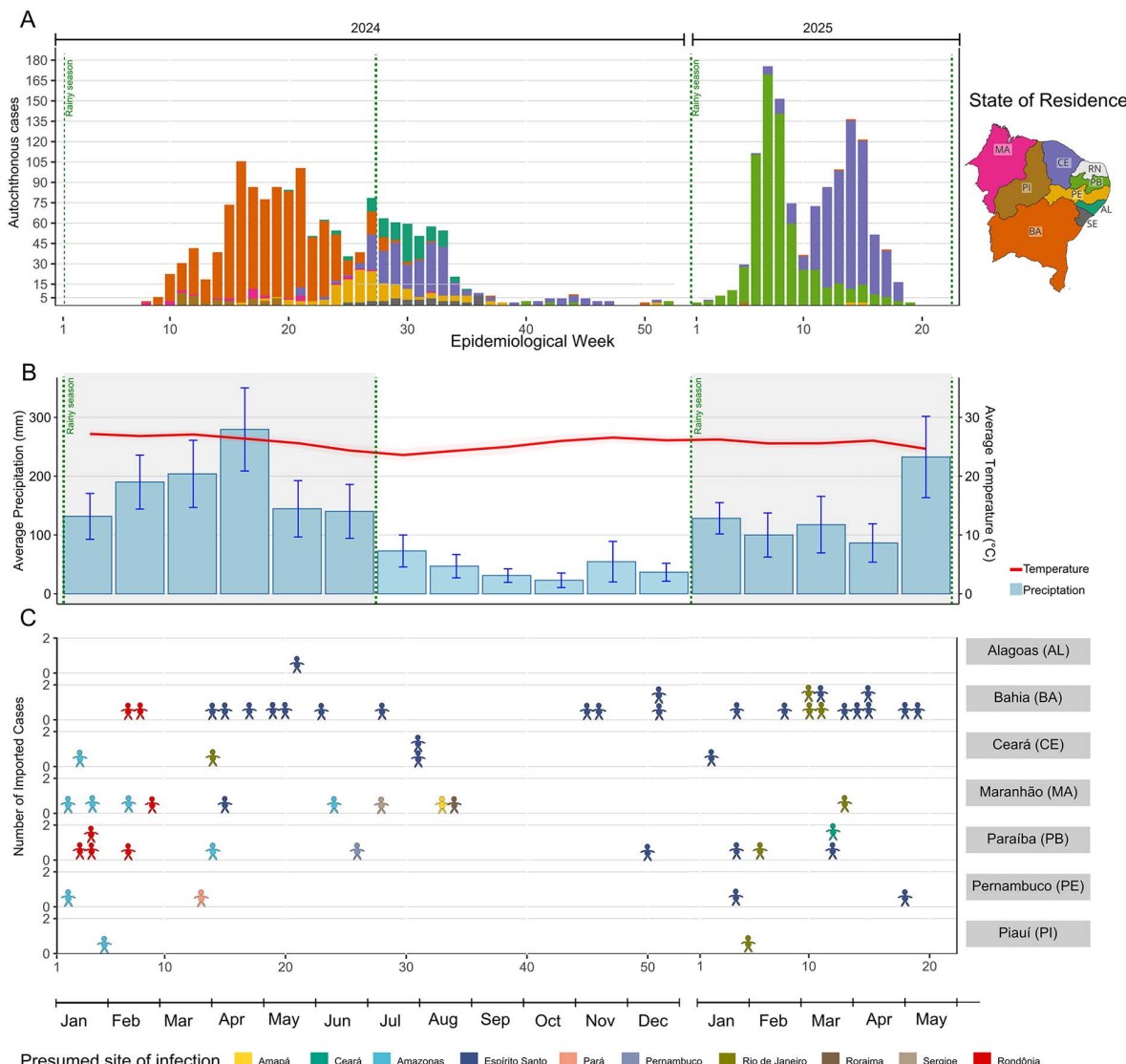

**Fig 2. Temporal distribution of autochthonous and imported OROV cases in Northeast Brazil by epidemiological week (2024–2025).** (A) Weekly distribution of autochthonous OROV cases by state of residence. The coloured bars highlight how the timing and magnitude of outbreaks varied across different states. (B) Average monthly precipitation (blue bars) and temperature (red line) in the areas reporting OROV cases in Northeast Brazil between January 2024 and May 2025. Error bars represent 95% confidence intervals in monthly precipitation. The shaded area highlights the rainy season. (C) Weekly distribution of imported OROV cases (cases in which the presumed site of infection differed from the municipality of residence), according to the state of residence (y-axis) and the presumed site of infection (color-coded). The basemap shapefiles used to produce Fig 2A were obtained from the IBGE municipal mesh dataset (available at: https://www.ibge.gov.br/en/geosciences/territorial-organization/territorial-meshes/2786-np-munici-pal-mesh/18890-municipal-mesh.html?lang=en-GB), which is distributed under a CC BY 4.0 license (https://biblioteca.ibge.gov.br/visualizacao/livros/liv102169.pdf).

## Municipal patterns of oropouche fever incidence in Northeastern Brazil

In Northeast Brazil, Oropouche fever cases were reported in 170 municipalities, representing 9.84% of all municipalities in the region. The spatial distribution was heterogeneous across states, with low municipal coverage within each state (Fig 3A). The lowest municipal coverage was observed in Piauí (2.77% of municipalities), while the highest occurred in

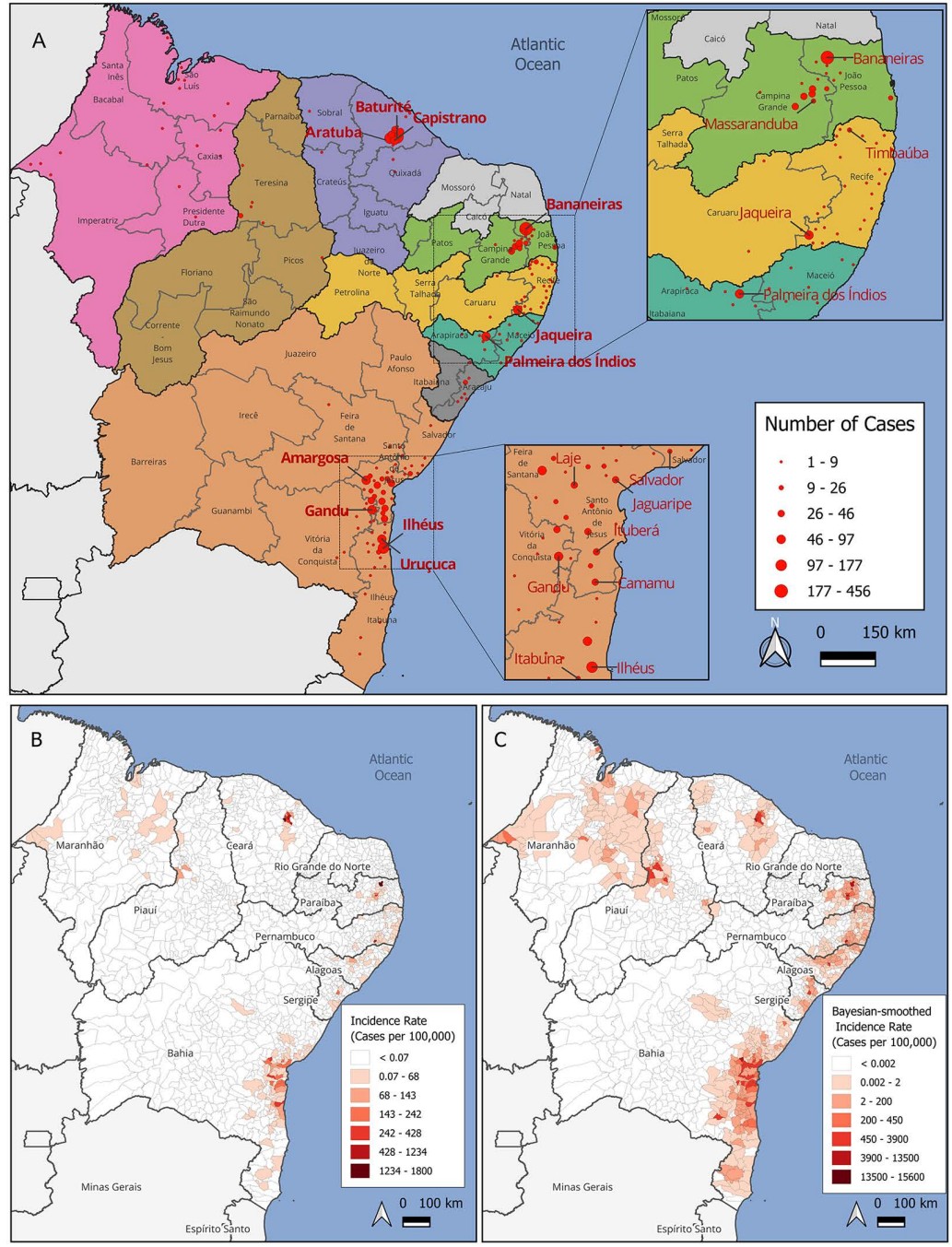

**Fig 3. Spatial distribution and incidence rates of OROV cases in Northeast Brazil.** (A) Absolute number of reported OROV cases per municipality between March 2024 and April 2025. The map is subdivided by states (colour-filled shapes) and Intermediate Geographic Regions (IGRs). Red circles represent the centroid of each municipality with reported cases. Labelled cities correspond to those with more than 46 cases in the main map and more than 10 cases in the inset maps. (B) Crude incidence rate of OROV cases per 100,000 inhabitants at the municipal level. (C) Bayesian-smoothed incidence rate per 100,000 inhabitants. The basemap shapefiles used to produce this figure were obtained from the IBGE municipal mesh dataset, available at: https://www.ibge.gov.br/en/geosciences/territorial-organization/territorial-meshes/2786-np-municipal-mesh/18890-municipal-mesh.html?lang=en-GB under a CC BY 4.0 license (https://biblioteca.ibge.gov.br/visualizacao/livros/liv102169.pdf).

Pernambuco (15.68%) (Table 1). Despite the higher proportion of affected municipalities in Pernambuco, the highest state-level incidence was recorded in Paraíba, with 15.54 cases per 100,000 population. In addition to these differences in municipal coverage and state-level incidence, a marked intra-state heterogeneity in municipal incidence rates was observed. In all states analyzed, municipal incidence coefficients of variation (CV) were high, indicating that transmission was concentrated in a few municipalities, while most reported low case numbers.

Most municipalities reporting OROV cases exhibited relatively low incidence rates (Fig 3B). Notably, however, a cluster of neighboring municipalities in northern Ceará, together with Jaqueira (Pernambuco), Matinhas and Bananeiras (Paraíba), and Uruçuca and Elísio Medrado (Bahia), displayed incidence rates exceeding 300 cases per 100,000 inhabitants (see S1 File - **Supplementary Results** for a detailed description). All these municipalities are relatively small in terms of population size.

Given this pattern, we examined heterogeneity across population sizes by grouping municipalities into population strata. Among municipalities with at least one reported case, the distribution of OROV incidence differed across population strata (Kruskal–Wallis $\chi^2 = 72.73$, df $= 2$, $p < 2.2 \times 10^{-16}$; $\varepsilon^2 = 0.424$) (S2 Fig). Median incidence was higher in small municipalities (median 17.3; IQR 7.47–67.6) than in medium-sized (median 1.80; IQR 1.16–7.91) and large municipalities (median 0.457; IQR 0.294–0.976). These comparisons are descriptive and limited to municipalities with at least one reported case. To formally test this relationship for all municipalities while accounting for excess zeros counts (i.e., many municipalities with no reported cases), we fitted a zero-inflated negative binomial model using population as an offset. Consistent with the stratified descriptive results, population size was inversely associated with incidence among municipalities with at least one reported case, with higher incidence tending to occur in smaller municipalities. Specifically, doubling the population size was associated with an approximately 46% reduction in estimated incidence (IRR $\approx 0.54$; $\beta = -0.879$; $p = 8.4 \times 10^{-7}$).

To further characterize the spatial distribution of OROV risk, we examined spatially smoothed incidence estimates. Spatial empirical Bayes smoothing revealed underlying spatial risk patterns (Fig 3C) that were not apparent in crude incidence maps, particularly in small municipalities with zero or very few reported cases. High-incidence areas remained prominent; however, spatial smoothing revealed that most municipalities near the coastal areas of Bahia, Sergipe, Alagoas, and Pernambuco present some degree of risk for Oropouche Fever. Additional risk was highlighted in municipalities within the Campina Grande Intermediate Geographic Region-IGR (Paraíba state) and along the Maranhão–Piauí border, where small populations had previously masked the underlying risk. This spatial pattern may reflect unrecognized transmission in

**Table 1. State-level distribution, incidence rates, and spatial heterogeneity of Oropouche fever in Northeast Brazil.**

| State | Number of municipalities | % of municipalities with cases[1] | State-level incidence (cases per 100,000 inhabitants) | Highest municipal incidence | Mean incidence[2] | Empirical Bayes incidence mean[3] | Coefficient of variation (%)[4] |
|---|---|---|---|---|---|---|---|
| Maranhão | 217 | 8.75 | 0.52 | 45.6 | 9.45 | 17.83 | 344 |
| Alagoas | 102 | 13.72 | 3.73 | 131.8 | 20.16 | 28.75 | 485 |
| Bahia | 417 | 15.11 | 6.03 | 329.8 | 56.30 | 132.0 | 335 |
| Ceará | 184 | 8.15 | 8.98 | 1544.64 | 271.78 | 589.4 | 358 |
| Paraíba | 223 | 7.62 | 15.54 | 1800.82 | 192.37 | 449.4 | 473 |
| Pernambuco | 184 | 15.68 | 1.54 | 648.67 | 31.79 | 90.0 | 738 |
| Piauí | 224 | 2.67 | 0.92 | 94.1 | 41.27 | 114.86 | 342 |
| Sergipe | 75 | 9.33 | 1.44 | 225.45 | 36.71 | 123.66 | 454 |

[1]Proportion of municipalities with at least one confirmed case of the disease.

[2]Mean of municipal incidence rates, considering only municipalities with at least one case.

[3]Mean of municipal incidence rates smoothed using the Spatial Empirical Bayes method, considering only municipalities with at least one case.

[4]Calculated using Empirical Bayes smoothed rates to minimize overestimation of heterogeneity due to small population sizes.

adjacent regions where outbreaks have been detected. The names of municipalities with smoothed incidence higher than 1000 are highlighted in S3 Fig.

We used spatial autocorrelation methods to identify whether municipalities with similar incidence were geographically clustered. A weak but statistically significant spatial autocorrelation was detected (Moran's I = 0.158; pseudo p-value = 0.002), suggesting a weak tendency for municipalities with similar incidence rates to be geographically clustered. The local spatial autocorrelation analysis (LISA) further identified four clusters of high disease incidence (S4 Fig; see S1 File - **Supplementary Results** for a detailed description).

## Ecological and climatic determinants of OROV case distribution

Oropouche fever cases occurred across municipalities located within all four major biomes present in Northeastern Brazil: the Amazon Forest, Cerrado, Caatinga, and the Atlantic Forest (Fig 4A). However, the distribution of cases was highly irregular. In 2024, the cases were predominantly reported in municipalities within the Atlantic Forest, following a trend observed in Brazil [13]. However, a notable shift was observed in 2025, when approximately 98% of all confirmed cases in the Northeast region were located within the Caatinga biome. This shift represents a substantial change in the ecological profile of Oropouche fever transmission, which, prior to 2023, had been largely confined to the Amazon Rainforest biome. Moreover, most OROV cases and all four spatial autocorrelation clusters with a large number of cases are restricted to per-humid, humid, and sub-humid climate zones. Therefore, even when the cases occur within the Caatinga biome, high-risk areas remain restricted to more humid climate zones and do not extend into the semi-arid region. Additionally, no cases were reported in areas experiencing more than nine dry months per year (S5A Fig). For reference, all climatic zones and precipitation regimes across Brazil are illustrated in S6 Fig.

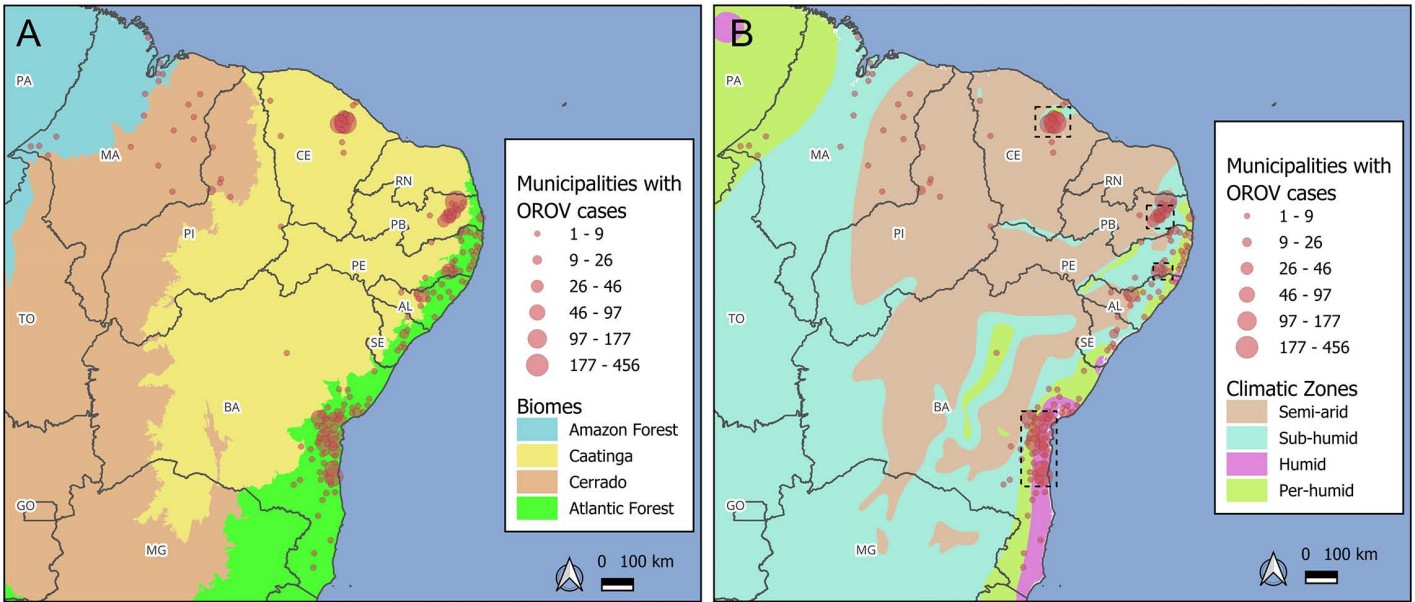

**Fig 4. Distribution of OROV cases by biomes (A) and climatic zones (B) in Northeast Brazil.** The maps show only cases reported within the Northeast region. Dashed-line rectangles in panel B indicate the four spatial autocorrelation clusters with a high number of cases, as identified by local Moran's I analysis. The basemap shapefiles used to produce this figure were obtained from the IBGE Brazilian terrestrial biomes dataset (available at: https://geoftp.ibge.gov.br/informacoes_ambientais/estudos_ambientais/biomas/vetores/Biomas_250mil.zip) and the IBGE climate dataset (available at: http://geoftp.ibge.gov.br/informacoes_ambientais/climatologia/vetores/brasil/Clima_5000mil.zip), both distributed under a CC BY 4.0 license (https://biblioteca.ibge.gov.br/visualizacao/livros/liv102169.pdf).

OROV cases were also reported across municipalities with diverse land use and land cover patterns, ranging from areas with extensive forest vegetation, through regions classified as mosaics of occupations in forest areas, agricultural zones, and even artificial (urban) areas (S5B Fig). In Pernambuco, for example, approximately 35% of cases had a probable site of infection in urban areas. No preferential vegetation pattern was observed among the municipalities with the highest OROV incidence in the Northeast region (S7 Fig). While some cities, such as Jaqueira (PE) and Amargosa (BA), are predominantly covered by agricultural and livestock lands, others, such as Aratuba (CE) and Baturité (CE), are mainly covered by primary vegetation. Conversely, cities like Gandu (BA) and Uruçuca (BA) are largely covered by secondary vegetation, whereas cities like Capistrano and Bananeiras are covered by a mix of savanna, submontane ombrophilous forest and agricultural lands. Among five high-incidence municipalities dominated by agricultural land, each from a different state, the most cultivated crops were banana in Bananeiras (PB, 44% of planted area), sugarcane in Jaqueira (PE, 69%), corn in Capistrano (CE, 58%), cocoa in Amargosa (BA, 43%), and both corn and beans in Palmeira dos Índios (AL, 32% each).

Genomic sampling and phylogenetic characterization of OROV from Pernambuco, Paraíba, and Sergipe

To improve understanding of the Oropouche fever epidemic in Northeast Brazil, we generated 65 near-complete OROV genomes from samples collected between March 2024 and April 2025 from three Northeastern states. This dataset included samples from Pernambuco (48 cases from 10 municipalities), Paraíba (17 cases from 8 municipalities), and Sergipe (1 case from 1 municipality). We then combined these genomes with previously published sequences from municipalities in Ceará [15] and Bahia [25], covering five of the eight Northeastern states that reported OROV cases during this period. Finally, we incorporated genomes from OROV cases reported in ten other Brazilian states in 2024, representing different regions of the country. The resulting alignment is available in S2 File.

Genotypic analysis confirmed that all 65 sequenced OROV genomes belonged to the $M_1L_2S_2$ lineage, which spread across multiple regions of Brazil in 2024. Without evidence of reassortment, the three genomic segments were concatenated, yielding a moderate temporal signal ($R = 0.81$, $R^2 = 0.64$, $p < 2.2 \times 10^{-16}$; S8 Fig).

Bayesian phylogenetic analysis revealed multiple independent introduction events of OROV lineages associated with outbreaks in Northeast Brazil, as well as inter-state viral dissemination within the region. We identified two distinct introductions into the state of Pernambuco (PE). The first, designated the **PE-I clade**, was directly derived from municipalities in the central region of Amazonas state (AM) (posterior probability [PP] = 100%) and was specifically linked to the **AM-I clade** (Fig 5A). The second introduction, referred to as the **PE-II clade**, originated in Rio de Janeiro (RJ) state (PP = 99%), following the prior establishment of the **AMACRO-II clade** in that region. Although Lima et al. (2025) [15] reported that the introduction of OROV into Ceará state also originated from the state of Amazonas, and despite the limited shared border between Pernambuco and Ceará, border, our Bayesian phylogenetic analysis indicates that the strains circulating in both states belongs to the same AM-I clade but represent two independent introduction and dissemination events into Northeast Brazil, with no interstate transmission between them (Fig 5A). Furthermore, our analysis shows that the introduction of the virus into Bahia, although derived from the **AMACRO-II clade**, represents a separate introduction event, distinct from the introduction event that gave rise to the **PE-II clade** (Fig 5A).

The estimated time to the most recent common ancestor (tMRCA) indicated that the **PE-I clade** likely emerged around February 2024 (95% highest posterior density [HPD]: December 2023–March 2024), preceding the emergence of the **CE clade**, estimated around March 2024 (95% HPD: January–May 2024), and the **PE-II clade**, which was dated to approximately May 2024 (95% HPD: March–June 2024) (Fig 5B). These estimates align with the temporal pattern of case detection shown in Fig 2A, suggesting that the virus circulated undetected for approximately two months before the onset of the recognized outbreak.

The PE-I clade exhibited the largest expansion within Pernambuco state and subsequently spread to neighboring states (Fig 5A). The first dispersal event was from the municipality of Jaqueira to Sergipe state (PP = 98%). This was followed by a separate introduction into Paraíba state, originating from Jaqueira and possibly reaching the municipality

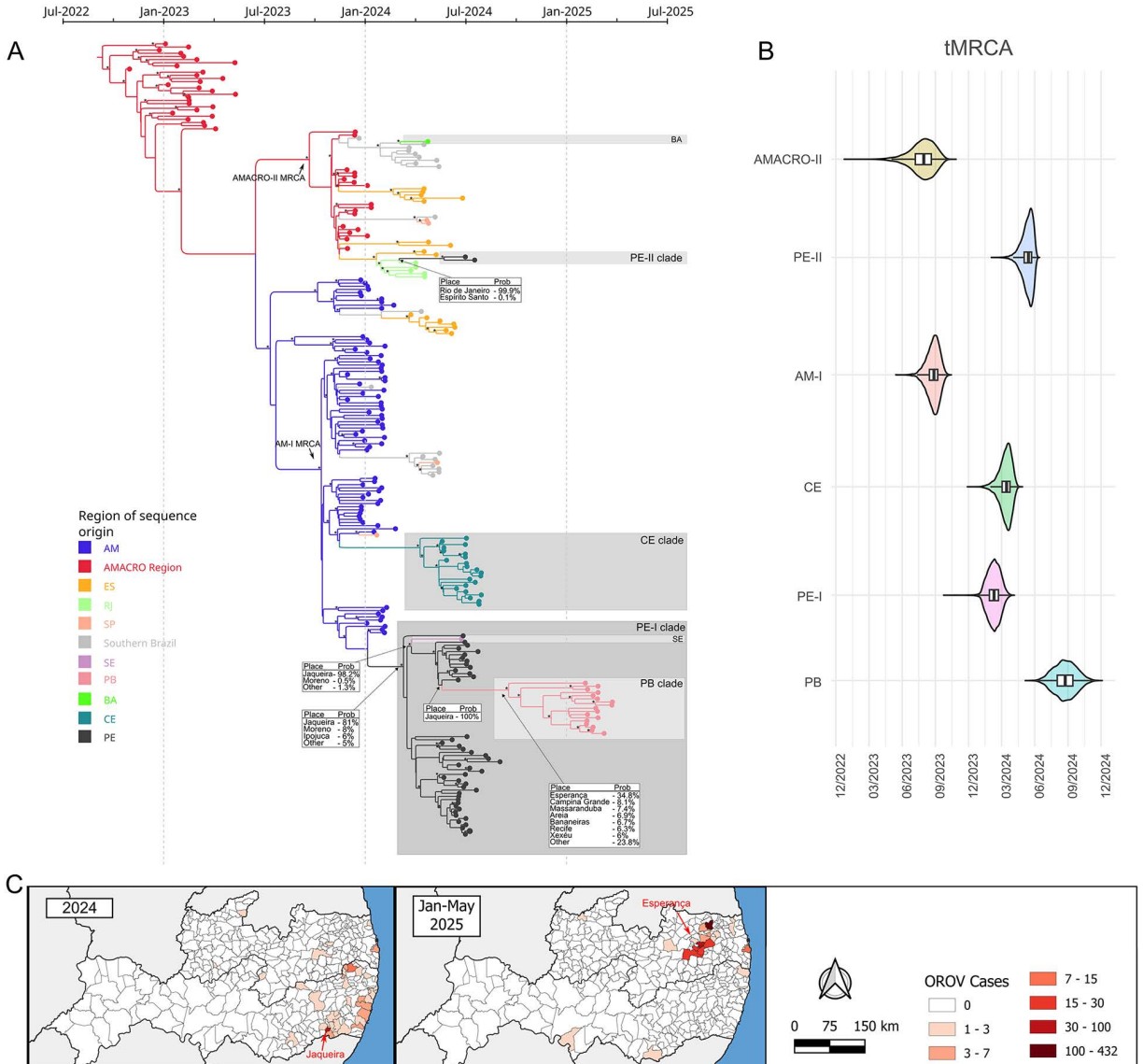

**Fig 5. Phylogenetic and temporal dynamics of OROV clades in Northeast Brazil (2024–2025).** (A) Time-scaled Bayesian phylogeny of OROV sequences from Brazil. Branches are color-coded by region of origin. AM = central/northern Amazonas, the other follows the same nomenclature of Fig 1. The AMACRO Region includes AC, RO, and southern Amazonas. Southern Brazil includes PR, SC, and RS. Nodes with posterior probabilities >0.9 are marked with an asterisk (*). Northeast Brazil samples are highlighted in grey-scale squares. Tables pointing to branches represent the probabilities of ancestor origin place (B) Estimated times to the most recent common ancestors (tMRCAs) for key regional clades. Violin plots show 95% highest posterior density (HPD) intervals. (C) Geographic distribution of OROV cases (imported and autochthonous) in Pernambuco and Paraíba. The municipalities of Jaqueira and Esperança are highlighted as initial points of viral dissemination in each period. The basemap shapefiles used to produce Fig 5C were obtained from the IBGE municipal mesh dataset (available at: https://www.ibge.gov.br/en/geosciences/territorial-organization/territorial-meshes/2786-np-municipal-mesh/18890-municipal-mesh.html?lang=en-GB), which is distributed under a CC BY 4.0 license (https://biblioteca.ibge.gov.br/visualizacao/livros/liv102169.pdf).

of Esperança (PP = 34.8%). Notably, all genomes sampled from municipalities with autochthonous cases in Paraíba clustered within a single, well-supported clade (PP > 0.9), suggesting that the recent outbreak in Paraíba was primarily driven by transmission of the PE-I clade. The phylogenetic model closely reflects the observed epidemiological patterns

of OROV case distribution across both states (Fig 5C). Taken together, these findings point to Jaqueira as the primary hotspot of OROV transmission in Pernambuco and the likely epicentre of regional viral spread.

To illustrate the spatio-temporal progression of OROV spread across Brazil, with particular focus on the Northeast region, we reconstructed a simplified phylogeographic model highlighting the lineages that gave rise to the outbreaks in Northeast Brazil (Fig 6A). Within Pernambuco, phylogeographic analyses revealed a sequential intra-state spread, primarily driven by the PE-I clade, originating in the Zona da Mata Sul region (part of the Atlantic Forest biome), specifically from the municipality of Jaqueira, which acted as the epicentre for this lineage within the state (Fig 6B). The first inferred transmission from Jaqueira reached nearby coastal municipalities, including Moreno, Ipojuca, Cabo de Santo Agostinho, and Recife. During their establishment in the southern region of the state, viruses from the PE-I Clade caused fetal deaths as reported by Medeiros et al. (2025) [7]. From the coast, the virus continued its expansion toward the Zona da Mata Norte, affecting municipalities such as Aliança, São Vicente Férrer, and Timbaúba. This last city also received a second introduction of the PE-II clade from RJ. In parallel with these movements, OROV also spread from Jaqueira to other neighboring municipalities and crossed state borders to reach Sergipe—the first state within the region where cases resulting from local (intra-regional) transmission were detected. All these movements occurred between May and September 2024, reflecting a period of intense viral movement and geographic expansion.

Jaqueira also served as a key source of viral spread into Paraíba state, with the first introduction reaching Esperança, according to our phylogeographic model (Fig 6C). From there, the virus spread to nearby municipalities, including Bananeiras, Areia, Massaranduba, and the state capital, João Pessoa. Bananeiras emerged as a major outbreak hotspot, concentrating a huge number of cases and likely driving further transmission to neighboring areas. These findings illustrate how rapidly OROV expanded and formed new transmission networks in regions where it had not circulated before.

## Discussion

This study characterized the rapid expansion of Oropouche virus across northeastern Brazil, a region that, until recently, had been largely unaffected by this pathogen. Between March 2024 and April 2025, both imported and local cases were detected, with sustained local outbreaks established in eight of the nine states, indicating the emergence of new potential endemic areas. The abrupt increase in case numbers, characterized by successive epidemic peaks within a short timeframe, underscore the virus' capacity to adapt to diverse epidemiological and ecological contexts. Although most cases across Northeastern Brazil are concentrated in the rainy season, the average precipitation levels are only about 50% of the rainfall observed in the state of Amazonas during the onset of the current OROV epidemic, between 2023 and 2024 [12]. This expansion has probably been driven by interstate population mobility and the presence of competent vectors in previously unaffected areas.

These patterns suggest an ongoing ecological diversification process, as OROV expanded into biomes and regions previously considered unsuitable for sustained transmission. Historically endemic to the Amazon region [36] and later detected in Atlantic Forest areas in 2024 [13], OROV began to show significant transmission in the Caatinga biome by 2025. The spatial spread of the currently circulating viral lineage, from the Amazon up to 2023, into the Atlantic Forest in 2024, and reaching the Caatinga in 2025, was supported by phylogeographic analyses, which revealed multiple introduction pathways from both Amazon Basin and Atlantic Forest regions. The Caatinga, a biome unique to Brazil and variably classified as a steppe savanna or dry tropical forest, is characterized by a predominantly semi-arid climate and pronounced intra-annual variability in precipitation [37]. This ecological profile contrasts sharply with the humid tropical environments of the Amazon, which span northern Brazil and neighboring countries such as Peru, Colombia, and Venezuela, nations that also reported OROV cases in 2025 [38]. However, Caatinga exhibits high ecoclimatic heterogeneity, and OROV cases were not uniformly distributed; instead, cases were concentrated in more humid subregions featuring patches of ombrophilous forest or arboreal or forested savanna, and areas with shorter dry seasons.

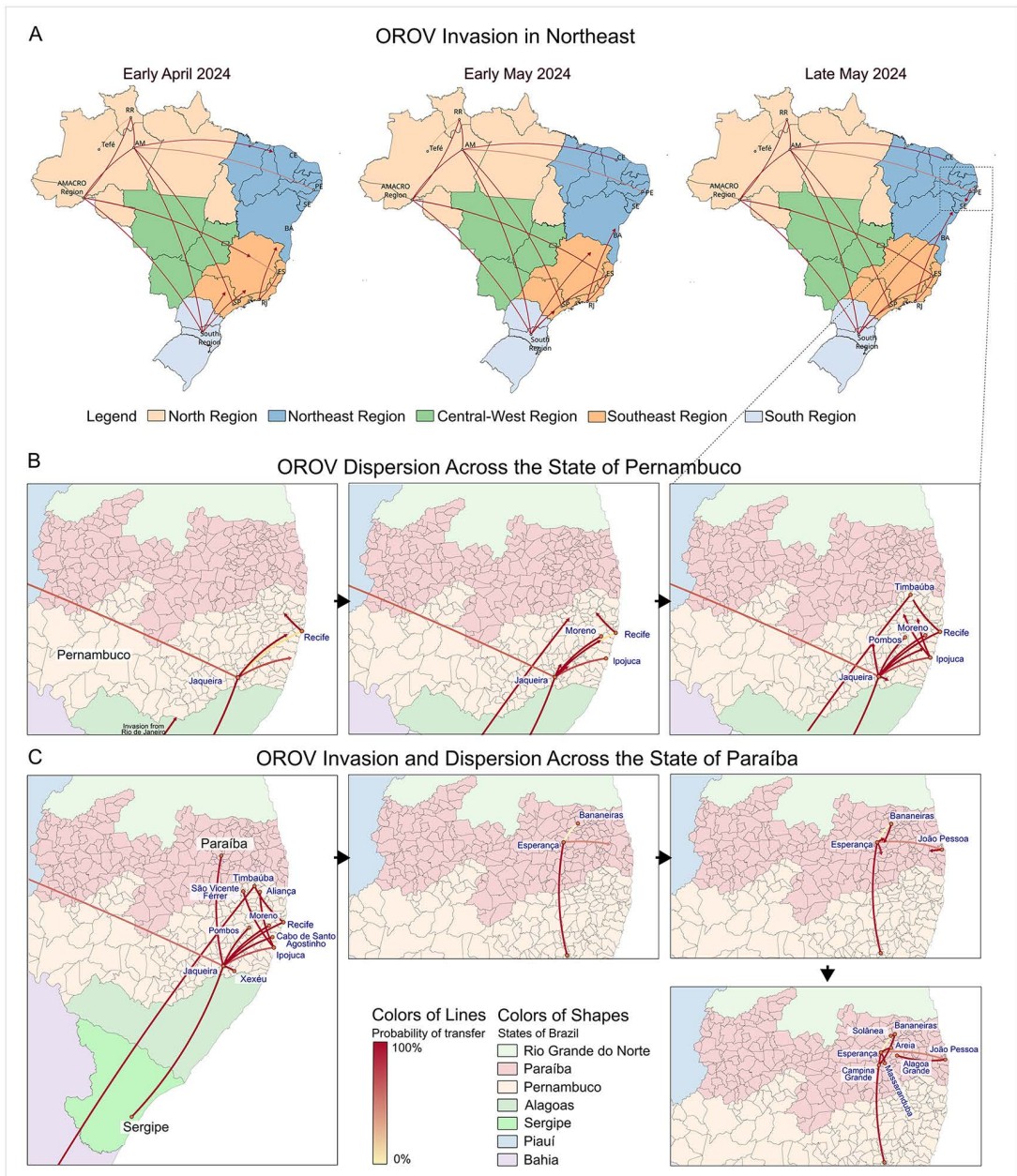

**Fig 6. Spatial and temporal dynamics of Oropouche virus (OROV) spread across Northeast Brazil.** (A) Spatiotemporal reconstruction of OROV introductions from other Brazilian regions into Northeast Brazil during April and May 2024. Maps show inferred viral movements over time, with arrows coloured according to the posterior probability of each transition. (B) Inferred dispersal routes among municipalities within Pernambuco state during 2024. (C) OROV invasion in Sergipe followed by subsequent introduction and spread within Paraíba state during late 2024 and early 2025. The base-map shapefiles used to produce this figure were obtained from the IBGE municipal mesh dataset (available at: https://www.ibge.gov.br/en/geosciences/territorial-organization/territorial-meshes/2786-np-municipal-mesh/18890-municipal-mesh.html?lang=en-GB), which is distributed under a CC BY 4.0 license (https://biblioteca.ibge.gov.br/visualizacao/livros/liv102169.pdf).

Humid and semi-humid environments favour the presence of *Culicoides paraensis*, a species that reaches high densities during the rainy season [39]. Although widely distributed across the Americas, from the United States to Uruguay [40], in northeastern Brazil, this vector has been recorded only in Bahia, Ceará, Maranhão, and Pernambuco [41]. This contrasts with recent outbreaks in states such as Piauí, Paraíba, Alagoas, and Sergipe, where no prior entomological records of the vector exist, highlighting critical gaps in entomological surveillance, as well as the possible involvement of alternative vectors, a hypothesis also raised in Venezuela [42]. While no other *Culicoides* species have been confirmed as OROV vectors, over 150 species of the genus are present in Brazil [41]. A study in Ceará [43] reported high species diversity near outbreak areas, including several anthropophilic taxa. Similar patterns were observed in municipalities bordering affected regions in Maranhão [44], with species that persist even during the dry season. Regarding mosquitoes, multiple species have been evaluated for their susceptibility to Oropouche virus (OROV) infection. Species such as *Ae. aegypti*, *Aedes scapularis*, *Ae. serratus*, *Cx. tarsalis*, and *Psorophora ferox* can become infected but appear unable to transmit the virus. In contrast, findings for *Culex quinquefasciatus* and *Aedes albopictus* are inconsistent, with most studies reporting no transmission and a few others indicating very low vector competence for viral transmission, which was evaluated using the first isolated OROV genotype [45]. Despite this, a recent study indicates that classical urban arbovirus vectors like *Ae. aegypti* and *Cx. quinquefasciatus* are refractory to the currently circulating OROV reassortment lineage under laboratory conditions [46], limiting their role in viral spread. Furthermore, if classical urban arbovirus vectors were substantially involved in OROV transmission, a much higher number of cases would be expected in cities that report high incidence of other arboviral diseases, such as dengue virus infection [45]. Considering these findings, enhanced entomological and molecular surveillance is urgently needed in states with confirmed OROV cases but no records of *C. paraenesis.* This is essential to address knowledge gaps and identify potential alternative vectors.

Previous records indicate that the Northeast region of Brazil had already been occasionally affected by Oropouche fever. Between 1987 and 1988, a localized outbreak occurred in Porto Franco (Maranhão), within the Cerrado biome [47]. In 2016–2017, five autochthonous cases were reported in Salvador (Bahia), but the origin of the virus remained unclear due to the absence of phylogeographic analyses and limitations in surveillance, including the lack of detection of imported cases [48]. Although underreporting remains a significant challenge, particularly outside the Amazon region, the recent spread of OROV in Northeast cannot be attributed to the expansion of testing following the viral emergence in 2023. Genomic sequencing performed on samples from three states, Pernambuco, Paraíba, and Sergipe, along with genomes from Ceará [15], revealed multiple independent viral introductions in 2024. These events were associated with long-distance human migrations, primarily from Amazonas and Rio de Janeiro, both of which had active transmission during that period. Following the introduction, the virus spread locally, likely driven by routine intermunicipal movement related to work, education, and healthcare, and eventually reached neighboring states, as observed in transmission flows between Pernambuco, Sergipe, and Paraíba. The genetic sequences revealed state-specific clustering and no evidence of cross-state reintroductions, suggesting that viral circulation remained predominantly intra-state after the initial introductions. Subsequently, the disease has been established and has become more prevalent in smaller municipalities across the Northeast, particularly those located farther from major urban centers. This pattern may be associated with the presence of forested or tree-covered areas, as observed during the initial phases of OROV expansion beyond the Amazon biome in 2024 [13].

An average interval of approximately two months was observed between the estimated date of viral introduction in a given locality (tMRCA) and the subsequent surge in case numbers, a pattern similar to that described in other OROV introductions in Brazil [13]. This interval may reflect the time required for infected individuals to return to their home states after traveling and initiate local transmission via vectors, combined with the viral replication period in insects and the human incubation period, which ranges from 1 to 15 days [49]. Additionally, recent data suggest the presence of replication-competent virus in the blood of patients for up to two months following symptom onset [50,51], potentially extending the period of infectivity to vector species, a hypothesis that remains to be rigorously tested through

experimental studies. The OROV tMRCA for Paraíba dates back to August/September 2024, and although few sporadic cases were recorded in the last three months of this year, the rapid growth in cases detected occurred only about 4 months later, which suggests that OROV circulated silently in Paraíba in the 2024 dry season after the wave of cases in Pernambuco had dissipated. Early testing and timely genomic sequencing are therefore critical to identify initial cases in a locality, facilitating not only the implementation of containment measures during this critical period but also the identification of viral introduction routes to inform targeted mitigation strategies.

The findings of this study highlight the urgent need to strengthen surveillance and response strategies, considering the expansion of Oropouche virus into regions previously considered non-endemic, but which already face challenges caused by other arboviruses such as DENV, ZIKV and CHIKV. The spatial smoothing approach applied in this analysis may help mitigate the effects of underreporting, yielding more robust estimates of disease risk. The results identify priority areas for syndromic surveillance of acute febrile illness, including nearly the entire coastal regions of Bahia and Pernambuco, as well as the states of Alagoas and Sergipe, in addition to specific clusters located in the interior of Paraíba and Ceará. Together with the observed lag between inferred viral introductions and subsequent case surges, these findings underscore the importance of improving early detection and routine surveillance of acute febrile illness, mainly in small municipalities of risk areas, which typically have weaker health surveillance structures. Lastly, population-based serological surveys in affected municipalities and neighboring areas could help quantify prior exposure and better characterize the extent of silent transmission suggested by our phylogeographical analyses and assess the risk of future outbreaks based on antibody prevalence.

This study has a number of limitations. Given that Oropouche fever has historically been underreported in many parts of Brazil and presents symptoms that resemble those of other arboviral infections, it is likely that numerous cases were not identified by local health systems, particularly in small municipalities with limited diagnostic capacity and healthcare infrastructure. To partially address this bias, we applied a spatial smoothing approach, which provided more robust estimates of disease risk across different locations in the Northeast region. However, it is likely that several patients were asymptomatic, and municipal hubs may also be underestimated. This limitation may have led to an underestimation of priority areas for future surveillance. Additionally, it was not possible to obtain viral genomes from all affected municipalities in the states of Pernambuco, Paraíba, and Sergipe. The phylogeographic model also relied on genome sequences previously generated by other research groups for Bahia and Ceará, and did not include data from Maranhão and Piauí. This may have led to an underrepresentation of viral introduction and dispersal routes across the broader Northeast region.

## Conclusion

This study provides robust evidence of the rapid and sustained transmission of Oropouche virus (OROV) across Northeast Brazil, driven by multiple independent introductions and local amplification networks. By integrating high-resolution genomic data with advanced spatial analyses, we identified critical dispersal hubs, such as the municipality of Jaqueira in Pernambuco, which played a central role in facilitating widespread regional dissemination. The combined analytical approach demonstrates how population mobility, favorable ecological niches, and immunologically naive populations synergistically contributed to the expansion of OROV into newly emergent endemic zones. These findings underscore the urgent need to enhance decentralized molecular diagnostic capacity, strengthen entomological and syndromic surveillance systems, and invest in healthcare workforce training across newly affected regions. The continued integration of genomic epidemiology and spatial analysis, as exemplified in this study, offers a powerful framework for anticipating and mitigating future outbreaks through more targeted and effective public health interventions.

## Supporting information

**S1 File. Supplementary results.** Municipal Patterns of Oropouche Fever Incidence in Northeastern Brazil.
(DOCX)

**S2 File. Multiple sequence alignment (FASTA format) of Oropouche virus (OROV) genomes from this study and previously published Brazilian sequences (2024–2025).** Sequences generated in this study were prefixed according to the following naming convention: hOROV/Brazil/PE-IAM, hOROV/Brazil/PE-LACENPE, hOROV/Brazil/PB-IAM, and hOROV/Brazil/SE-IAM.
(FA)

**S1 Fig. Weekly distribution of autochthonous and imported OROV cases by state of residence, between 2024 and 2025.**
(JPG)

**S2 Fig. Distribution of Oropouche fever incidence (log10 scale) by municipality size (≤50,000; 50,000–200,000; and >200,000 people) in Northeast Brazil.** Each point represents a municipality. Boxplots show the median and the interquartile range (IQR). Differences between groups were tested using the Kruskal–Wallis test ($p < 2.2 \times 10^{-16}$). Brackets indicate all pairwise comparisons, with their p-values. Numeric values are shown on the x-axis, but the axis is on a logarithmic scale.
(PNG)

**S3 Fig. Smoothed incidence rates of Oropouche virus by state in Northeast Brazil.** Each violin plot represents the distribution across municipalities whose smoothed incidence is greater than zero. The names of municipalities with smoothed incidence higher than 1000 are highlighted.
(JPG)

**S4 Fig. Local Indicators of Spatial Association (LISA) for Empirical Bayes smoothed incidence rates of Oropouche virus in Northeast Brazil.** Top panel: Cluster map showing spatial autocorrelation patterns. Bottom panel: LISA significance map indicating municipalities with statistically significant local spatial autocorrelation at $p < 0.05$, 0.01, and 0.001. The basemap shapefiles used to produce this figure were obtained from the IBGE municipal mesh dataset (available at: https://geoftp.ibge.gov.br/organizacao_do_territorio/malhas_territoriais/malhas_municipais/municipio_2023/Brasil/BR_Municipios_2023.zip), which is distributed under a CC BY 4.0 license (https://biblioteca.ibge.gov.br/visualizacao/livros/liv102169.pdf)
(JPG)

**S5 Fig. Distribution of municipalities with OROV cases across precipitation regime zones (A) and land cover and use (B) in Northeast Brazil.** The basemap shapefiles used to produce this figure were obtained from the IBGE climate dataset (available at: http://geoftp.ibge.gov.br/informacoes_ambientais/climatologia/vetores/brasil/Clima_5000mil.zip) and the IBGE land cover and use dataset (available at: https://www.ibge.gov.br/geociencias/downloads-geociencias.html?-caminho=informacoes_ambientais/cobertura_e_uso_da_terra/monitoramento/grade_estatistica/serie_revisada_2022/vetores_compactados/UFs, both distributed under a CC BY 4.0 license (https://biblioteca.ibge.gov.br/visualizacao/livros/liv102169.pdf)
(JPG)

**S6 Fig. Climatic zones and precipitation regimes across Brazil.** The basemap shapefiles used to produce this figure were obtained from IBGE climate dataset (available at: http://geoftp.ibge.gov.br/informacoes_ambientais/climatologia/vetores/brasil/Clima_5000mil.zip), which is distributed under a CC BY 4.0 license (https://biblioteca.ibge.gov.br/visualizacao/livros/liv102169.pdf)
(JPG)

**S7 Fig. Vegetation characteristics of municipalities with high incidence of Oropouche virus in five northeastern Brazilian states.** Each panel compares vegetation cover (left) and land use classification from MapBiomas 2023 (right),

highlighting the presence of forest fragments, transitional zones, and agricultural areas within affected municipalities. The basemap shapefiles used to produce this figure were obtained from the IBGE Brazilian vegetation dataset (available at: https://geoftp.ibge.gov.br/informacoes_ambientais/vegetacao/vetores/escala_250_mil/versao_2023/) and the MapBiomas Collection 2023, which provides an annual series of land cover and land use maps of Brazil (accessed via the official Map-Biomas Collection plugin for QGIS: https://github.com/mariochermes/mapbiomascollection), distributed under a CC BY 4.0 license (https://brasil.mapbiomas.org/en/termos-de-uso/).
(PNG)

**S8 Fig. Root-to-tip divergence regression of OROV genomes showing a moderate temporal signal (R = 0.81).**
(JPG)

## Acknowledgments

We thank the FIOCRUZ Genomic Surveillance Network for their continued support, the Brazilian Ministry of Health for providing access to case data through its official portal, the public health and medical teams across Brazil for their response to the Oropouche outbreak, and the Bioinformatics Center of the Aggeu Magalhães Institute for their support.

## Author contributions

**Conceptualization:** Elverson Soares de Melo, Clarice Neuenschwander Lins de Morais, Gabriel da Luz Wallau.

**Formal analysis:** Elverson Soares de Melo, Sophia Maria Dantas da Silva, Gabriel da Luz Wallau.

**Investigation:** Elverson Soares de Melo, Sophia Maria Dantas da Silva, Gustavo Barbosa de Lima, Alexandre Freitas da Silva, Verônica Gomes da Silva, Elisa de Almeida Neves Azevedo, Letícia Welter Rother, Keilla Maria Paz e Silva, Diego Arruda Falcão, Andreza Pâmela Vasconcelos, Mayara Matias de Oliveira Marques da Costa, Eduardo Augusto Duque Bezerra, Thiago Franco de Oliveira Carneiro, Erik Matthaus de Lima Paiva, Janaina Correia Oliveira, Matheus Filguera Bezerra, Marcelo Henrique Santos Paiva, Bartolomeu Acioli-Santos, Clarice Neuenschwander Lins de Morais, Tulio de Lima Campos, Gabriel da Luz Wallau.

**Methodology:** Elverson Soares de Melo, Sophia Maria Dantas da Silva, Adalúcia da Silva, Gabriel da Luz Wallau.

**Resources:** Marcelo Henrique Santos Paiva, Bartolomeu Acioli-Santos, Clarice Neuenschwander Lins de Morais, Tulio de Lima Campos, Gabriel da Luz Wallau.

**Supervision:** Gabriel da Luz Wallau.

**Visualization:** Elverson Soares de Melo, Sophia Maria Dantas da Silva.

**Writing – original draft:** Elverson Soares de Melo, Sophia Maria Dantas da Silva, Bartolomeu Acioli-Santos, Clarice Neuenschwander Lins de Morais, Gabriel da Luz Wallau.

**Writing – review & editing:** Elverson Soares de Melo, Adalúcia da Silva, Marcelo Henrique Santos Paiva, Tulio de Lima Campos, Gabriel da Luz Wallau.

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
