## [Decision Letter · Decision Letter 0]

10 Dec 2025

Oropouche Virus Outbreaks in Northeast Brazil between 2024-25 are characterized by Sustained Transmission and Spread to Newly Affected Areas

Dear Dr. Wallau,

Thank you for submitting your manuscript to PLOS Neglected Tropical Diseases. After careful consideration, we feel that it has merit but does not fully meet PLOS Neglected Tropical Diseases's publication criteria as it currently stands. Therefore, we invite you to submit a revised version of the manuscript that addresses the points raised during the review process.

We look forward to receiving your revised manuscript.

Kind regards,

Abdallah Samy

Section Editor

Shaden Kamhawi

co-Editor-in-Chief

Paul Brindley

co-Editor-in-Chief

**Additional Editor Comments :**

Thank you for considering PLOS Neglected Tropical Diseases for your submission. I kindly ask that you address all concerns raised by the reviewers before we can consider a revised version of your manuscript.

**Journal Requirements:**

At this stage, the following Authors/Authors require contributions: Elverson Soares de Melo, Sophia Maria Dantas da Silva, Gustavo Barbosa de Lima, Adalucia da Silva, Alexandre Freitas da Silva, Veronica Gomes da Silva, Elisa de Almeida Neves Azevedo, Leticia Welter Rother, Keilla Maria Paz e Silva, Diego Arruda Falcao, Andreza Pamela Vasconcelos, Mayara Matias de Oliveira Marques da Costa, Eduardo Augusto Duque Bezerra, Thiago Franco de Oliveira Carneiro, Erik Matthaus de Lima Paiva, Janaina Correia Oliveira, Matheus Filguera Bezerra, Marcelo Henrique Santos Paiva, Bartolomeu Acioli-Santos, Clarice Neuenschwander Lins de Morais, Tulio de Lima Campos, and Gabriel Luz Wallau. Please ensure that the full contributions of each author are acknowledged in the "Add/Edit/Remove Authors" section of our submission form.

Potential Copyright Issues:

i) Figure 2C. Please confirm whether you drew the images / clip-art within the figure panels by hand. If you did not draw the images, please provide (a) a link to the source of the images or icons and their license / terms of use; or (b) written permission from the copyright holder to publish the images or icons under our CC BY 4.0 license. Alternatively, you may replace the images with open source alternatives. See these open source resources you may use to replace images / clip-art:

ii) Figures 1, 2A, 3, 4, 5C, 6, S4, S5, S6, and S7. Please (a) provide a direct link to the base layer of the map (i.e., the country or region border shape) and ensure this is also included in the figure legend; and (b) provide a link to the terms of use / license information for the base layer image or shapefile. We cannot publish proprietary or copyrighted maps (e.g. Google Maps, Mapquest) and the terms of use for your map base layer must be compatible with our CC BY 4.0 license.

3) If any authors received a salary from any of your funders, please state which authors and which funders.

7) Please revise your current Competing Interest statement to the standard "The authors have declared that no competing interests exist."

**Reviewers' Comments:**

**Comments to the Authors:**

**Please note that one review is uploaded as an attachment.**

Reviewer's Responses to Questions

**Key Review Criteria Required for Acceptance?**

**Methods**

-Are the objectives of the study clearly articulated with a clear testable hypothesis stated?

-Is the study design appropriate to address the stated objectives?

-Is the population clearly described and appropriate for the hypothesis being tested?

-Is the sample size sufficient to ensure adequate power to address the hypothesis being tested?

-Were correct statistical analysis used to support conclusions?

-Are there concerns about ethical or regulatory requirements being met?

Reviewer #1: Please see Summary and General Comments section

Reviewer #2: (No Response)

Reviewer #3: The detailed comments are in the attached Word document.

**Results**

-Does the analysis presented match the analysis plan?

-Are the results clearly and completely presented?

-Are the figures (Tables, Images) of sufficient quality for clarity?

Reviewer #1: Please see Summary and General Comments section

Reviewer #2: (No Response)

Reviewer #3: The detailed comments are in the attached Word document.

**Conclusions**

-Are the conclusions supported by the data presented?

-Are the limitations of analysis clearly described?

-Do the authors discuss how these data can be helpful to advance our understanding of the topic under study?

-Is public health relevance addressed?

Reviewer #1: Please see Summary and General Comments section

Reviewer #2: (No Response)

Reviewer #3: The detailed comments are in the attached Word document.

**Editorial and Data Presentation Modifications?**

Reviewer #1: n/a

Reviewer #2: (No Response)

Reviewer #3: Minor Revision

**Summary and General Comments**

Reviewer #1: This is an important and impactful report on OROV surveillance and its current distribution in Brazil. The work is comprehensively presented, methods are sound and the figures are quite informative. Information contained in the manuscript is of seminal importance to the field.

That said, I suggest the authors to address the following concerns prior an eventual publication.

1. The statistical statements seem to contrast with the actual numbers. The authors claim "A correlation analysis between population size and the smoothed incidence rate across municipalities in the Northeast region of Brazil revealed a moderate negative association in the log-transformed space (r = −0.36; p < 2.2×10−16), indicating..." It seems a stretch to draw any conclusion from a Pearson correlation coefficient of 0.36 (positive or negative). Likewise, from a 'weak' autocorrelation.

2. The authors state that "[...] age distribution, most cases (62.19%) occurred among the 20 to 49 age group." What about the other nearly 40%. A bar or violin plot of the age distribution would be more informative.

3. There is a section in the discussion that is not well connected to the manuscript findings. It is rather structured as proposal pitch and does not contribute to the work. I suggest to remove entirely or dramatically restructure the following sentences: "Mapping areas at risk can guide targeted investment in the continuous training of healthcare professionals for the recognition and clinical management of suspected OROV cases, while also supporting the expansion of access to decentralized molecular testing. Strengthening local diagnostic capacity, including improvements in sample storage and RNA extraction procedures, can minimize viral RNA degradation, improve specimen quality for genomic sequencing, and reduce underreporting. Investment in rapid detection systems is particularly critical given the potential for viral reemergence in municipalities or regions that did not experience major outbreaks but possess climatic and ecological conditions favourable to Culicoides spp. proliferation. In such settings, the low recorded incidence is likely insufficient to establish herd immunity. Lastly, randomized seroepidemiological surveys in affected municipalities are recommended to more accurately estimate prior population-level exposure and, consequently, to assess the risk of future outbreaks based on antibody prevalence."

Reviewer #2: The manuscript entitled “Oropouche Virus Outbreaks in Northeast Brazil between 2024–25 Are Characterized by Sustained Transmission and Spread to Newly Affected Areas” provides a comprehensive characterization of the introduction and spread of OROV in Northeast Brazil during the current outbreak. The study encompasses epidemiological, genomic, and ecological aspects, offering an integrated perspective on the dynamics of OROV transmission in the region.

The manuscript is clear and well written, and the authors provide a detailed description of the methods applied. The methodological approaches are appropriate, and the results are thoroughly and thoughtfully discussed.

The results presented by the authors are highly important for public health, as they provide critical insights into the dynamics of Oropouche virus transmission, support timely detection of outbreaks, and help guide surveillance and control strategies in newly affected regions. I just bring few points to be revised in the manuscript before publication:

1- “Individuals from Pernambuco, Paraíba, Ceará, Maranhão, and Piauí likely acquired the infection while traveling to the Amazonian region (Fig. 2C).”

Please clarify how the authors inferred these likely travel-associated infections in the sentence above, and also the presumed site of infection in Figure 2C. Was this conclusion based on phylogenetic reconstruction, epidemiological information, or another data source? If the inference relied on some epidemiological databases (e.g., SINAN), please reference this use in the Methods section.

2- “Additional risk was highlighted in municipalities within the Campina Grande IR (Paraíba)...”

Please, revise this sentence and provide the meaning or context of “IR”.

3- “The findings of this study highlight the urgent need to strengthen surveillance and

response strategies, considering the expansion of Oropouche virus into regions previously

considered non-endemic, but which already face challenges caused by other arboviruses such

as DENV, ZIKV and CHIKV.”

Please, include the abbreviations for these three viruses, along their full names, at their first citation in the manuscript.

Reviewer #3: The detailed comments are in the attached Word document.

PLOS authors have the option to publish the peer review history of their article (what does this mean?). If published, this will include your full peer review and any attached files.). If published, this will include your full peer review and any attached files.). If published, this will include your full peer review and any attached files.). If published, this will include your full peer review and any attached files.

...

Reviewer #1: No

Reviewer #2: **Yes:** Richard Steiner SalvatoRichard Steiner SalvatoRichard Steiner SalvatoRichard Steiner Salvato

Reviewer #3: No

**Figure resubmission:**
---

## [Decision Letter · Decision Letter 1]

18 Mar 2026

Dear Dr. Wallau,

We are pleased to inform you that your manuscript 'Oropouche Virus Outbreaks in Northeast Brazil between 2024-25 are characterized by Sustained Transmission and Spread to Newly Affected Areas' has been provisionally accepted for publication in PLOS Neglected Tropical Diseases.

Best regards,

Abdallah M. Samy, PhD

Section Editor

Abdallah Samy

Section Editor

Shaden Kamhawi

co-Editor-in-Chief

Paul Brindley

co-Editor-in-Chief

Reviewer's Responses to Questions

**Key Review Criteria Required for Acceptance?**

**Methods**

-Are the objectives of the study clearly articulated with a clear testable hypothesis stated?

-Is the study design appropriate to address the stated objectives?

-Is the population clearly described and appropriate for the hypothesis being tested?

-Is the sample size sufficient to ensure adequate power to address the hypothesis being tested?

-Were correct statistical analysis used to support conclusions?

-Are there concerns about ethical or regulatory requirements being met?

Reviewer #1: (No Response)

Reviewer #2: (No Response)

Reviewer #3: (No Response)

**Results**

-Does the analysis presented match the analysis plan?

-Are the results clearly and completely presented?

-Are the figures (Tables, Images) of sufficient quality for clarity?

Reviewer #1: (No Response)

Reviewer #2: (No Response)

Reviewer #3: (No Response)

**Conclusions**

-Are the conclusions supported by the data presented?

-Are the limitations of analysis clearly described?

-Do the authors discuss how these data can be helpful to advance our understanding of the topic under study?

-Is public health relevance addressed?

Reviewer #1: (No Response)

Reviewer #2: (No Response)

Reviewer #3: (No Response)

**Editorial and Data Presentation Modifications?**

Reviewer #1: (No Response)

Reviewer #2: (No Response)

Reviewer #3: (No Response)

**Summary and General Comments**

Reviewer #1: The authors have satisfactorily addressed all concerns in the revised version of the manuscript.

Reviewer #2: (No Response)

Reviewer #3: The authors have adequately addressed all of my comments. I am satisfied with the revisions, and I would recommend accepting this manuscript for publication.

PLOS authors have the option to publish the peer review history of their article (what does this mean?). If published, this will include your full peer review and any attached files.). If published, this will include your full peer review and any attached files.). If published, this will include your full peer review and any attached files.). If published, this will include your full peer review and any attached files.

...

Reviewer #1: No

Reviewer #2: No

Reviewer #3: No

---

## [Editor Report · Acceptance letter]

Dear Dr. Wallau,

We are delighted to inform you that your manuscript, "Oropouche Virus Outbreaks in Northeast Brazil Between 2024–2025 Are Characterized by Sustained Transmission and Spread to Newly Affected Areas.," has been formally accepted for publication in PLOS Neglected Tropical Diseases.

Best regards,

Shaden Kamhawi

co-Editor-in-Chief

Paul Brindley

co-Editor-in-Chief
